



# Anthropogenic Reversal of the Natural Ozone Gradient between Northern and Southern Mid-latitudes

David D. Parrish[1], Richard G. Derwent[2], Steven T. Turnock[3], Fiona M. O'Connor[3], Johannes Staehelin[4], Susanne E. Bauer[5,6], Makoto Deushi[7], Naga Oshima[7], Kostas Tsigaridis[6,5], Tongwen Wu[8] and Jie Zhang[8]

[1]David.D.Parrish, LLC, Boulder, CO, 80303, USA
[2]rdscientific, Newbury, Berkshire, RG14 6LH, United Kingdom
[3]Met Office Hadley Centre, Exeter, UK
[4]ETH Zurich, Zurich, Switzerland
[5]NASA Goddard Institute for Space Studies, New York, NY, USA
[6]Center for Climate Systems Research, Columbia University, New York, NY, USA
[7]Meteorological Research Institute, 1-1 Nagamine, Tsukuba, Ibaraki, 305-0052, Japan
[8]Beijing Climate Center, China Meteorological Administration, Beijing, China

*Correspondence to*: David D. Parrish (david.d.parrish.llc@gmail.com)

**Abstract.** Our quantitative understanding of natural tropospheric ozone concentrations is limited by the paucity of reliable
measurements before the 1980s. We utilize the existing measurements to compare the long-term ozone changes that occurred
within the marine boundary layer at northern and southern mid-latitudes. Since 1950 ozone concentrations have increased by
a factor of $2.1 \pm 0.2$ in the northern hemisphere (NH) and are presently larger than in the southern hemisphere (SH), where
only a much smaller increase has occurred. These changes are attributed to increased ozone production driven by anthropogenic
emissions of photochemical ozone precursors that increased with industrial development. The greater ozone concentrations
and increases in the NH are consistent with the predominant location of anthropogenic emission sources in that hemisphere.
The available measurements indicate that this interhemispheric gradient was much smaller, and was likely reversed in the
natural troposphere with higher concentrations in the SH. Six Earth System Model (ESM) simulations indicate similar total
NH increases (1.9 with a standard deviation of 0.3), but they occurred more slowly over a longer time period, and the ESMs
do not find higher pre-industrial ozone in the SH. Several uncertainties in the ESMs may cause these model-measurement
disagreements: the assumed natural nitrogen oxide emissions may be too large, the relatively greater fraction of ozone injected
by stratosphere-troposphere exchange to the NH may be overestimated, ozone surface deposition to ocean and land surfaces
may not be accurately simulated, and model treatment of emissions of biogenic hydrocarbons and their photochemistry may
not be adequate.

## 1 Introduction

Ozone ($O_3$) is a species of central importance to tropospheric chemistry. Foremost, it is the primary precursor of the hydroxyl
radical (Levy, 1971), which drives much of tropospheric photochemistry. This photochemistry oxidizes many air pollutants
(e.g., carbon monoxide, hydrocarbons, oxides of nitrogen and sulfur dioxide, among others) yielding less toxic (e.g., carbon



dioxide and water) or more soluble species (e.g., nitric and sulfuric acids) that precipitation rapidly removes from the atmosphere. Thus, the hydroxyl radical, and thereby its ozone precursor, effectively cleans the atmosphere. However, ozone

also has harmful environmental impacts; it is an air pollutant that adversely affects human, crop and ecosystem health, and it acts as a greenhouse gas, thus contributing to climate change (Monks et al., 2015). A comprehensive understanding of the distribution of ozone concentrations and ozone sources and sinks, both in time and in space, is needed for formulating effective policies for regulating ozone concentrations.

Ozone has both natural and anthropogenic (i.e., photochemical air pollution) sources that are balanced by deposition and in

situ chemical loss processes. The magnitude of these sources and sinks vary widely throughout the troposphere. In regions relatively isolated from photochemical precursor emissions, production and loss rates are slow compared to atmospheric transport. Thus, ozone concentrations are the product of local sources and sinks, modulated by transport of ozone-rich or ozone-poor air from other regions of the troposphere. Simulating these complex and interrelated photochemical, physical and transport processes is challenging, and significant shortcomings in global chemical transport model results have been

identified, including in simulations of long-term changes (e.g., Staehelin et al., 2017 and references therein) and seasonal cycles (Parrish et al., 2016).

Published analyses of long-term changes in tropospheric ozone have an inconsistency that has not been widely discussed. Parrish et al. (2012, 2014) find that tropospheric ozone concentrations increased by about a factor of 2 between 1950 and 2000 at mid-latitudes in the northern hemisphere (NH). However, comparisons of ozone concentrations between hemispheres (e.g.,

Figure 1 of Cooper et al., 2014) indicate that ozone has changed little in the southern hemisphere (SH), and that present-day ozone is higher in the NH, but by a factor of less than 2. For example, Derwent et al. (2016) report mean ozone mixing ratios of $38.9 \pm 0.4$ ppb (1989-2014) and $32.0 \pm 0.7$ ppb (1990-2010) at two NH marine boundary layer (MBL) sites, and $25.0 \pm 0.2$ ppb (1982-2010) at a comparable SH site. A simple hypothesis can resolve this inconsistency - before the natural ozone distribution was perturbed by anthropogenic emissions of ozone precursors, the ozone gradient was reversed compared to that

of today, with concentrations higher in the SH than the NH at mid-latitudes. The primary goal of this paper is to test this hypothesis through comparison of measured long-term ozone changes at mid-latitudes in the two hemispheres, thereby quantifying the pre-industrial and present-day interhemispheric ozone gradients.

Global atmospheric chemistry model simulations indicate that a reversal of the interhemispheric ozone gradient is plausible. Wang and Jacob (1980) used a global three-dimensional model of tropospheric chemistry to investigate pre-industrial ozone

levels and discussed uncertainties and potential difficulties. Of particular importance was the level of the pre-industrial NOx emissions. Figure 1 examines the dependence of the interhemispheric ozone gradient upon the assumed magnitude of the natural NOx emissions through five simulations of the global Lagrangian chemistry-transport model (STOCHEM-CRI). The base case assumes no NOx emissions to the troposphere; under these conditions photochemical ozone production is small, with the required NOx precursor provided only by formation from nitric acid entering the troposphere through stratosphere-

troposphere exchange. A reversed ozone gradient arises from the faster surface deposition of ozone to land surfaces in the northern hemisphere. Biomass burning emissions in the second simulation contain a wide range of trace gases, but with NOx



emissions excluded, ozone levels increase somewhat but the reversed gradient is not significantly affected. As pre-industrial $NO_x$ emissions (from lightning, soil emissions, and biomass burning) are added step-wise (third through fifth simulations), ozone levels rise and the reversed gradient is gradually eroded at mid-latitudes.

Comparisons of observed ozone concentrations with simulations by modern global atmospheric chemistry models provides useful tests of the models, and hopefully useful guidance for their improvement. One fruitful comparison arises from analysis of measurement records to establish, as accurately as possible, the long-term ozone changes that occurred in the background troposphere as industrial development proceeded, particularly at northern mid-latitudes. Challenges for this analysis include the sparseness of the measurement record and uncertainty regarding the accuracy of measurements made by different
researchers using a variety of techniques at different locations. The Task Force of Hemispheric Transport of Air Pollutants (HTAP, 2010; Parrish et al., 2012; 2014) quantified long-term ozone changes at northern mid-latitude sites, predominately in Europe. As part of the Tropospheric Ozone Assessment Report (https://igacproject.org/activities/TOAR), Tarasick et al. (2019) critically reviewed the record of historical ozone measurements throughout the global troposphere. Parrish et al. (2020a) have recently synthesized the HTAP and TOAR analyses. A second goal of this work is to compare these observational analyses
with results from recent earth system model (ESM) simulations.

        In this work, the interhemispheric comparison of long-term tropospheric ozone changes is limited to the MBL, since the available measurements from the SH were all collected in the MBL; there are no higher elevation SH sites with extensive records. Nevertheless, long-term changes of ozone in the MBL do reflect the concurrent changes in the free troposphere, because entrainment of ozone from the free troposphere is the primary source of ozone to the MBL. Parrish et al. (2012; 2014;
2020b) demonstrate that long-term ozone changes are similar from the surface to the mid-troposphere based upon comparisons of observed concentrations at baseline-representative sites in the MBL, at mountain top sites, and in the free troposphere from balloon borne sondes and aircraft.

        High quality measurements from two MBL sites - Cape Grim, Australia in the SH and Mace Head, Ireland in the NH – provide the primary basis for our analysis. These measurements extend from the 1980s to the present and are selected for
baseline conditions, thus representing the background troposphere. (By baseline conditions, we mean measurements collected during periods when air is transported to the measurement site from the marine environment, thereby avoiding confounding influences from nearby continental areas - see discussion in Chapter 1 of HTAP, 2010). Two different strategies are employed to extend each of these ozone records back to 1950. Before the 1970s, limited MBL measurements are also available from other sites in both hemispheres that provide support for the ozone changes derived for the 1950 to 1980 period. These
measurements were conducted at coastal sites relatively isolated from nearby influences, so they are expected to be directly comparable to the two primary data sets. Post-1980 data from other MBL sites in both hemispheres provide comparisons for the more recent Cape Grim and Mace Head data.



## 2 Methods

The analysis in this paper is based on the results of published observational and model simulation studies that allow
quantification of long-term ozone changes from 1950 to the present in the MBL at mid-latitudes in both hemispheres. No
continuous ozone measurement record covering the entire period from pre-industrialization to the present exists in either
hemisphere; thus, we use less direct approaches to construct long-term ozone records from the available observations. Herein
we consistently express ozone concentrations as mole fractions (i.e., mixing ratios) in units of nmol $O_3$/mole air (referred to as
ppb). Quantitative results are given with indicated uncertainties, which are 95% confidence limits unless stated otherwise.

The analysis presented here is based on MBL ozone observations that fall into two categories: recent (1982 to 2017)
continuous measurements, and older (1956 to 1984) measurements made over limited time periods. Table 1 gives the locations,
elevation, and years of measurements for these data sets. The primary analysis is based on the continuous, recent measurements
from Mace Head and Cape Grim. The Mace Head data are those selected as representative of the unpolluted NH MBL (i.e.,
baseline conditions) as discussed by Derwent et al. (2018a); Table 1 in Appendix A of their Supplementary Data gives the
monthly means, which are considered here. The annual mean Cape Grim data were downloaded from the TOAR data archive
(Schultz et al., 2017; https://join.fz-juelich.de/access/, last accessed 20 April 2020); they were selected for baseline conditions
as described in the TOAR data header. Parrish et al. (2020b) discuss a time series of baseline selected, seasonal mean ozone
mixing ratios derived from measurements at sites in the Pacific MBL at the US west coast; these data provide a comparison
for the Mace Head data. The US Pacific MBL monthly mean data are given in Table S1 of the Supporting Information. For
the analysis here, annual means at Mace Head and the US Pacific MBL are derived from the tabulated monthly means for each
year with all 12 months of data available. Data from all times of day are included for the Cape Grim, Mace Head and US
Pacific MBL data sets.

The older measurements are multi-year means taken from Tarasick et al. (2019). We utilize the results accepted into their
record (their Tables 4 and 6) from mid-latitude MBL sites (included in Table 1) in both hemispheres. In figures showing the
results, the mean of the measurements made over multiple years are plotted at the center of the measurement period. Tarasick
et al. (2019) conclude that some of these results are of questionable reliability, a conclusion that is considered in the discussion
of these data. The TOAR effort did not attempt baseline filtering for these ozone records, so any impact of local or regional
pollution-related influences remains unquantified.

The measurement programs providing the two primary data sets were initiated too late (1982 and 1988 at Cape Grim and
Mace Head, respectively) to directly characterize ozone changes from 1950 to the present. Southern mid-latitude long-term
ozone changes have been small (e.g., Cooper et al., 2014; Tarasick et al., 2019), so a standard linear regression fit to the Cape
Grim annual means extrapolated back to 1950 is the most suitable method to derive an ozone trend over the entire period;
Table 2 gives the parameters of this fit. Long-term ozone changes at northern mid-latitudes have been much larger than in the
SH (e.g., Cooper et al., 2014), so a simple extrapolation approach is not appropriate; here we quantify the long-term ozone
change at Mace Head from the analysis developed during the HTAP study (HTAP, 2010; Parrish et al. 2012; 2014).



The HTAP analysis utilized the sparse record of early measurements made at baseline representative sites throughout Europe to quantify long-term ozone changes on that continent. These measurements extend back to 1950, with two summer measurement periods from the 1930s. The long-term ozone change quantification is based on relative ozone changes, which are derived by dividing each time series of seasonal means at each measurement site by the year 2000 intercept of a fit to those

means. Parrish et al. (2014) found that the relative long-term changes are the same within statistical confidence limits at all baseline-representative European sites, but with some seasonal differences. Later analysis (Parrish et al., 2020b) showed that those seasonal differences are not significant, and that the relative long-term ozone changes are the same, within statistical confidence limits, for all seasons. Figure 2a shows the 757 relative seasonal means available from all seasons at six baseline-representative European sites. Even though the relative ozone concentrations have substantial scatter, the large number of data

allow precise polynomial fits to the overall time series. The fits included in Figure 2a are a linear regression over 1950 to 2000, a cubic (i.e., $3^{rd}$ order) polynomial over 1950 to 2010, and a $4^{th}$ order polynomial over 1934-2010. The three fits give similar changes over the 1950 to 2000 period: factors of 2.23, 2.16 and 2.08 for the linear, cubic and $4^{th}$ order polynomial fits respectively. Each of these three factors agree with the 1950 to 2000 relative change of $2.1 \pm 0.2$ derived in a synthesis of the HTAP and TOAR analyses (Parrish et al., 2020a). Parrish et al. (2014) also show that simulations by three CMIP5 global

chemistry climate models agree that relative means in all seasons at all European baseline-representative sites exhibit similar relative long-term changes; simulations from one model are shown in Figure 2b and from the other two models in Figure S1. Figures S1-S8 of Parrish et al. (2014) illustrate the normalization process and analysis for these same measurement and model results for separate seasons. Figure 2 indicates that to estimate the long-term ozone change at Mace Head (or any other baseline representative site in western Europe), one needs only to quantify the year 2000 mean ozone at the site, and then calculate the

product of that intercept with the polynomial fit.

The European historical ozone data considered by Parrish et al. (2014) and included in Figure 2a lacked quantified uncertainties. However, the accuracy of relative long-term ozone changes derived from these data is supported by the critical evaluation of Tarasick et al. (2019), which found no significant, systematic inaccuracy in the historical data analyzed by Parrish et al. (2012; 2014). Parrish et al. (2020a; their Figure 1) show that the seasonal and annual mean long-term changes derived as

described above provide good fits to all of the historical European data identified by Tarasick et al. (2019). The observations in Figure 2a do show substantial scatter about the fits, with a root-mean-square deviation (RMSD) of 8.9% of the year 2000 intercept for the cubic fit. To be representative of the historical data, this RMSD must be referenced to the smaller magnitude of the historical data; the corresponding RMSD is then ~16% when referenced to the year 1960 intercept. Tarasick et al. (2019) estimate a relative uncertainty of 0.7-1.2 at approximately 90% confidence intervals for the methods employed to collect the

historical data; this corresponds to a relative standard deviation of ~15%, assuming a normal distribution of measurement errors. Thus, the historical data included in Figure 2a are judged to be as accurate and precise as can be expected from the historical measurement methods.

The model results that we compare to the measurements are from three sources: six ESMs (identified in Figure S2 and S3) that took part in the CMIP6 exercise, three CCMs (identified in Figure S4) that took part in the CMIP5 exercise, and the





STOCHEM-CRI model described by Derwent et al. (2018b). Table S2 references descriptions of the CMIP6 ESMs, and Parrish et al. (2014) give more details and references for the simulation results of the CMIP5 CCMs. Surface concentrations of ozone at Mace Head and Cape Grim were obtained from the six CMIP6 models that had made data available on the Earth System Grid Federation (ESGF) at the time of writing. Ozone concentrations were also obtained from these same models for the model level that included the site elevation for 8 additional NH and 2 additional SH baseline sites. All ESM results are from the coupled historical simulations over the 1850 to 2014 period from all available ensemble members of each CMIP6 model. STOCHEM-CRI is a global Lagrangian chemistry-transport model with a detailed description of tropospheric chemistry which makes it suitable for studies of low NOx and isoprene chemistry (Jenkin et al., 2019) and the pre-industrial atmosphere (Khan et al., 2015). STOCHEM-CRI is driven by meteorological fields from the UK Meteorological Office Unified Model taken from an archive for 1998, further details of which are given in Collins et al. (1997); ozone sources in this model are dominated by stratosphere-troposphere exchange, which was set to 745 Tg $O_3$ yr$^{-1}$.

## 3 Results

The sparse records of available baseline ozone measurements made in the mid-latitude MBL of both hemispheres are compared in Figure 3. The symbols after 1980 are annual means of baseline selected data from three representative long-term measurement records – Cape Grim in the SH and Mace Head and the U.S. Pacific MBL in the NH. The seven symbols before 1980, two in the SH and five in the NH, represent relatively short (2 to 16 years), year-round records evaluated in TOAR (Tarasick et al., 2019). The map in Figure 3 identifies the site locations, and Table 1 gives site and data record information. We quantify the MBL baseline ozone mixing ratios as accurately as possible from these limited data, to allow a comparison of long-term changes between the two hemispheres.

The year 2000 annual mean mixing ratios at Mace Head and the U.S. Pacific MBL are 39.8 ± 0.6 ppb and 32.9 ± 1.1 ppb, respectively (Parrish et al., 2020b). The difference in these annual means represents significant zonal variation in MBL ozone concentrations between the eastern North Pacific and eastern North Atlantic Oceans. Since the NH MBL data reported by Tarasick et al. (2019) are all from European measurements, we will primarily focus on the Mace Head data in this analysis. The Mace Head measurements agree closely with other North Atlantic MBL sites; year 2000 annual mean mixing ratios are 38.5 ± 0.5 and 37.3 ± 1.3 ppb at Storhofdi, Iceland and Tudor Hill, Bermuda, respectively (Parrish et al., 2016). The long-term trends derived for the two representative NH sites before 2010 (i.e., beginning in 1988) are consistent with each other: 0.31 ± 0.10 ppb yr$^{-1}$ at Mace Head, and 0.27 ± 0.08 ppb yr$^{-1}$ at the U.S. Pacific MBL (Cooper et al., 2014).

Baseline selected measurements at Cape Grim extend back to 1982. Parrish et al. (2016) show that these data agree closely with those at other southern mid-latitude MBL sites - year 2000 annual mean mixing ratios of 25.0 ± 0.2 ppb at Cape Grim compared to 23.1 ± 0.4 ppb and 23.7 ± 0.4 ppb at Cape Point, South Africa and Ushuaia, Argentina, respectively. These results from three continents demonstrate the zonal similarity of tropospheric ozone concentrations at southern mid-latitudes. Thus, we take the Cape Grim data record to be representative of the entire southern mid latitude MBL. The solid green line in Figure



2 indicates the linear fit over the entire data record, with a small, but statistically significant, long-term trend of $0.041 \pm 0.019$ ppb yr$^{-1}$, which is a factor of ~7 smaller than at the two NH sites. To guide the discussion of changes that may have occurred before the beginning of these measurements, this linear fit is extrapolated back to 1950. This extrapolation may overestimate

the earlier changes, as it is not known when the increase began. However, this trend is small enough that this uncertainty has negligible impact on the following discussion.

As is apparent from Figure 3, no single site measurement record covers the complete 1950 to 2010 period or includes annual means before 1950 at northern mid-latitudes. However, the quantification of the common relative long-term ozone change over western Europe from all available baseline sites (Figure 2) allows estimation of the long-term ozone change at Mace Head

for that period; the cubic fit (green curve) illustrated in Figure 2 multiplied by the Mace Head year 2000 mean ozone (given above) yields the brown polynomial curve in Figure 3 (coefficient values given in Table 2). The long-term changes in ozone derived for Mace Head are given by that curve with the shading indicating the confidence limits derived from the propagation of the confidence limits of the Mace Head year 2000 mean and the overall relative increase of baseline ozone over Europe of a factor of $2.1 \pm 0.2$ (Parrish et al., 2020a).

The means of the older (1960-1980) data included Figure 3 do not differ significantly between the two hemispheres; they are $23.5 \pm 1.8$ and $22.5 \pm 2.2$ ppb (where standard deviations are indicated) for the two SH and five NH data sets, respectively. These limited data indicate that the large interhemispheric gradient apparent in today's measurements was not present in the 1960-1980 period. Figure 3 shows that the extrapolation of the Cape Grim data agrees closely with the SH data from Macquarie Is. and Hermanus, in accord with the small trends and zonal similarity of tropospheric ozone concentrations at southern mid-

latitudes. The mean of the five NH points, all from European measurements, is smaller than the 27.2 ppb value of the brown curve in 1970, and this curve is above all of the pre-1980 MBL measurements as well as the more recent US Pacific MBL data; this indicates that this curve, derived for Mace Head, provides an upper limit for NH mid-latitude baseline ozone in the MBL. It should be noted that Tarasick et al. (2019) judge four (Norderney, Cagliari, Westerland and Hermanus) of these seven earlier data sets to be of questionable reliability; however, exclusion of these data does not significantly change the overall

agreement of the 1960-1980 data with the derived changes in either the NH or SH.

## 4 Discussion

The ozone measurements illustrated in Figure 3 lead to the conclusion that tropospheric ozone concentrations were higher in the SH than the NH before industrial development. Three lines of reasoning support this deduction. First, the sparse measurement record at baseline sites indicates that ozone concentrations in the NH increased by a factor of $2.1 \pm 0.2$ between

1950 and 2000 (Figure 2 and Parrish et al., 2020a), yet present NH ozone concentrations are less than a factor of 2 greater than those in the SH, indicating that ozone concentrations must necessarily have been lower in the NH than the SH in 1950. Second, the brown and green curves in Figure 3 intersect in about 1962, indicating that the NH ozone concentrations were lower than those in the SH in earlier years. Third, that curve intersection indicates that the MBL ozone concentrations were similar in both



hemispheres in the 1960s, and the few available measurements from that period are all consistent with that similarity; however,
multiple considerations indicate that anthropogenic precursor emissions had already substantially increased NH hemisphere
ozone concentrations by that time. By 1962 the brown curve in Figure 3 had increased by ~20% from 1950 levels, and global
model calculations (see discussion below) also find that NH ozone concentrations had increased significantly before the 1960s.
Further, extremely elevated ozone concentrations (several 100 ppb) were observed in Los Angeles as early as the 1950s
(Haagen-Smit, 1954). The emissions responsible for those urban ozone enhancements were primarily from on-road vehicles,
which were common to all U.S. urban areas. Those U.S. emissions, and similar emissions in other countries at northern mid-
latitudes, are expected to have impacts throughout the NH mid-latitude troposphere. Thus, although the measurement record
is sparse, observations and the wider considerations discussed above are all consistent with the conclusion that in the pre-
industrial troposphere, mid-latitude ozone concentrations were higher in the SH than the NH. The cause of the greater increases
in the NH and reversal of the natural interhemispheric ozone gradient is attributed to the increased emissions of ozone
precursors that accompanied industrial development, and those increased emissions were predominantly located in the NH.

Differences in ozone sinks and/or sources between hemispheres can account for larger natural ozone concentrations in the
SH compared to the NH. The loss rate of ozone to ocean surfaces is slow, but is much faster over continents due to surface
deposition to vegetation and to reaction with natural hydrocarbons emitted from forests. The fractional coverage of
midlatitudes by land is ~50% in the NH, but only 6 to 7% in the SH, implying significantly slower ozone losses in that
hemisphere. Molecular hydrogen may provide an analogy to ozone. Its average concentration is higher in the SH (Simmonds
et al., 2000); this distribution is attributed to greater uptake by soils in the NH, despite there being active photochemical sources
and sinks of hydrogen in both hemispheres and evidence for significant anthropogenic pollution sources concentrated in the
NH. However, for hydrogen the greater NH pollution source is not large enough to reverse the natural interhemispheric gradient
of hydrogen.

Comparisons between observations and global model simulations provide a basis to elucidate the causes of the changing
interhemispheric ozone gradient, as well as to assess model performance. Figure 4 compares the observational-derived long-
term changes from Figure 3 with time series of annual mean surface ozone at Mace Head and Cape Grim simulated by six
ESMs that participated in the 6th Coupled Model Intercomparison Project (CMIP6, Eyring et al., 2016). Figure S2 shows those
same model simulations for the entire 1850-2014 simulation period with each of the six models identified. In Figure 4 the
simulated concentrations in recent years (after ~1990) all agree within ~5 ppb with the observations in both hemispheres.
Section S1 of the Supporting Information gives further discussion of this agreement at 10 NH and 3 SH baseline sites, and
includes comparisons with simulations by three CMIP5 models.

The ESM simulations generally agree that long-term changes in SH ozone have been small, but they do not reproduce the
rapid increase in ozone that occurred in the NH between 1950 and 2000. At Cape Grim the mean model trend over the 1982-
2014 period of observations is 0.082 ppb yr$^{-1}$, with model results varying from 0 to 4 times the observed trend of $0.041 \pm 0.019$
ppb yr$^{-1}$. In a comparison of simulations by four of these same ESMs, Griffiths et al. (2020) found similarly good agreement
at Cape Grim, as well as three other remote background sites, but did not compare those simulations with any northern mid-





latitude observations. The modeled 1950 to 2000 increases at northern mid-latitudes correspond to a mean factor of 1.3 with a standard deviation of 0.1, while observations indicate a $2.1 \pm 0.2$ factor increase. It is notable that the overall modeled northern

mid-latitude ozone increase since pre-industrial times is a factor of 1.9 with a standard deviation of 0.3, as judged from the mean model ratio over the entire 1850 to 2000 period; this value agrees more closely with the observed 1950 to 2000 ratio of $2.1 \pm 0.2$, which we interpret as a good approximation for the factor of total ozone increase during northern mid-latitude industrialization. This closer agreement is in accord with the analysis of Staehelin et al. (2017) that also found smaller model simulated ozone increases at northern mid-latitudes over the post-1950 period. They suggest that this discrepancy may be

attributable to problems in quantifying changes in the historical emissions of ozone precursors from anthropogenic sources, and discuss evidence for a significant impact from such problems.

The ESMs also do not simulate lower pre-industrial mid-latitude ozone in the NH; the 1850 mean NH/SH ratio is 1.13 with a standard deviation of 0.11, with only one of the six ESMs finding a ratio (slightly) less than unity. The full range of the five pre-industrial STOCHEM-CRI scenarios (with natural NOx emissions between 0.5 and 18 Tg N yr$^{-1}$) from Figure 1 are also

indicated in Figure 4; the scenarios that gave the largest ozone concentrations, i.e., those with the largest pre-industrial NOx emissions, generally agree with the ESM simulations, but those with lower NOx emissions give lower ozone concentrations with SH ozone higher than NH ozone. The absence of a reversed ozone gradient in the ESM simulations may indicate that the assumed natural NOx emissions are too large. For reference, the natural NOx emissions assumed in three of the six ESMs considered here are ~11 to 14 Tg N yr$^{-1}$ (Figure 1 of Griffiths et al., 2020), which are near the larger of the pre-industrial

STOCHEM-CRI scenarios. If the natural NOx emissions are too large, then the model calculated radiative forcing of ozone is too small.

Other processes that affect ozone also differ significantly between hemispheres; thus, uncertainties in their parametrizations may also contribute to ESMs simulating higher pre-industrial ozone in the NH. Stratosphere-troposphere exchange (STE) is an important natural ozone source. A recent review of the tropospheric ozone budget (Archibald et al., 2020a) suggests that

the stronger Brewer-Dobson circulation in the NH produces a larger STE ozone flux in that hemisphere (~57% of total). However, Škerlak, et al., (2015) find that deep tropopause folds, which are most efficient for transporting stratospheric ozone into the lower troposphere, are more frequent in the SH. The absolute and relative magnitudes of ozone loss to land and ocean surfaces differ strongly between hemispheres. Luhar et al. (2018) give a new parameterization scheme that reduces deposition to oceanic surfaces by a factor of ~3 compared to earlier work; incorporation of this new result into ESMs would raise SH

ozone relative to the NH. The surface deposition of ozone to land surfaces occurs predominately in the NH, and its representation in current models is regarded as insufficient (Clifton et al., 2020). Another concern is the model treatment of the chemistry of natural hydrocarbons (e.g., isoprene and terpenes) emitted in large quantities from temperate forests that are predominately located in the NH. At the low NOx concentrations believed to have dominated the pre-industrial continental boundary layer, this chemistry constitutes an important ozone sink, while at the higher modern-day NOx concentrations, it is

an ozone source. Understanding the NOx concentration dependence of this complex natural hydrocarbon chemistry is still an



active area of research (e.g., Jenkin et al., 2015). The magnitude of the pre-industrial emissions of these natural hydrocarbons is also quite uncertain (Mickley et al., 2001), and the CMIP6 models use a variety of estimation approaches.

In addition to the above-discussed model uncertainties that most directly affect the ozone gradient, Wild et al. (2020) identify key areas in model simulations that require improvement for accurate simulation of the tropospheric ozone distribution; these

include the atmospheric water vapor distribution and the drivers of variability in global OH, which differ significantly between models. Derwent et al. (2020) identify key improvements required in the representation of the atmospheric chemistry of the pre-industrial troposphere in ESMs and other global chemistry-transport models. Additional improvements to the treatment of the atmospheric chemistry of natural and anthropogenic ozone precursors, especially NOx, and of ozone loss processes are likely required to accurately treat the balance of ozone production and loss in both the present day and pre-industrial

troposphere, a requirement necessary to accurately model the interhemispheric ozone gradient and fully understand the radiative forcing of tropospheric ozone.

**Acknowledgments**

The authors are grateful for discussions with Ian Galbally, Maria Val Martin, Simone Tilmes, Fred Fehsenfeld and Owen Cooper. P.G. Simmonds and T.G. Spain provided the Mace Head data and A.J. Manning sorted the Mace Head data into

baseline and non-baseline observations. Ian Galbally and Suzie Molloy provided the baseline selected Cape Grim data; and the work of the staff of Cape Grim are acknowledged. D.D.P. acknowledges support from NOAA's Atmospheric Chemistry and Climate Program. NOAA Global Monitoring Laboratory provided the Trinidad Head ozone and meteorology data. S.T.T. would like to acknowledge that support for his work came from the BEIS and DEFRA Met Office Hadley Centre Climate Programme (GA01101) and the UK-China Research and Innovation Partnership Fund through the Met Office Climate Science

for Service Partnership (CSSP) China as part of the Newton Fund. M.D. and N.O. were supported by the Japan Society for the Promotion of Science (grant numbers: JP18H03363, JP18H05292, and JP20K04070) and the Environment Research and Technology Development Fund (JPMEERF20172003, JPMEERF20202003, and JPMEERF20205001) of the Environmental Restoration and Conservation Agency of Japan, and the Arctic Challenge for Sustainability II (ArCS II), Program Grant Number JPMXD1420318865. S.E.B. and K.T. acknowledge resources supporting this work were provided by the NASA High-

End Computing (HEC) Program through the NASA Center for Climate Simulation (NCCS) at Goddard Space Flight Center. The CESM project is supported primarily by the National Science Foundation. Computing and data storage resources, including the Cheyenne supercomputer (doi:10.5065/D6RX99HX), were provided by the Computational and Information Systems Laboratory (CISL) at NCAR. NCAR is sponsored by the National Science Foundation. R.G.D. provided the model results from STOCHEM-CRI with help from Anwar Khan and Dudley Shallcross of the University of Bristol. Disclosure:

D.D.P. also works as an atmospheric chemistry consultant (David D. Parrish, LLC); he has had contracts funded by several state and federal agencies and an industrial coalition, although they did not support the work reported in this paper.



**Data Availability:**

All of the data utilized in this paper are available from public archives referenced in this paper, and from Table S1 of the Supporting Information.

**Author Contributions:**

D.D.P. and R.G.D. designed research and performed analysis; S.E.B., M.D., N.O., K.T., T.W., J.Z. and R.G.D. performed model simulations; S.T.T. extracted model simulation results; D.D.P. wrote the paper with input from all other authors.

**Competing interests:** The authors declare that they have no conflict of interest.

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




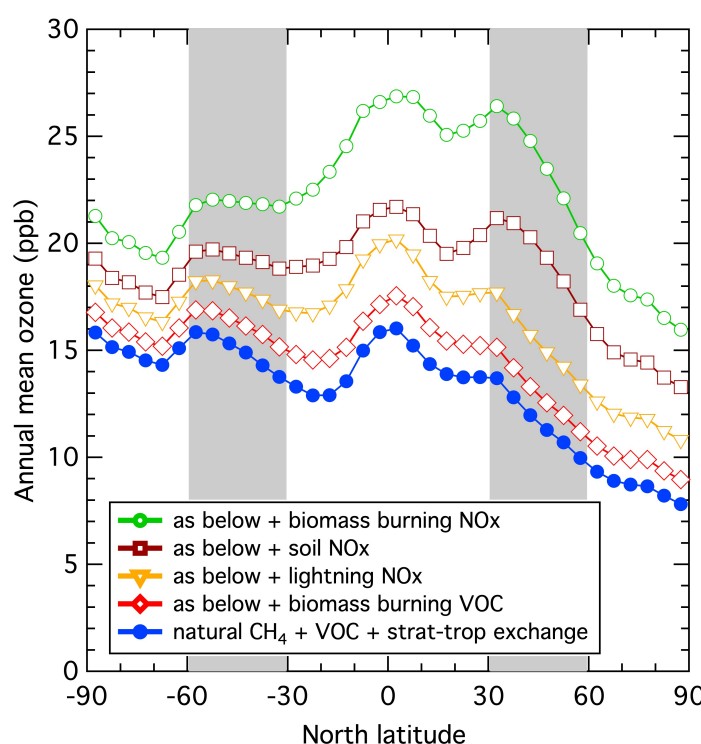

**Figure 1. Pre-industrial latitude dependence of zonal surface annual mean ozone mixing ratios calculated by the STOCHEM-CRI for progressively larger assumed NOx emission scenarios. The base case assumes a pre-industrial methane mixing ratio of 718 ppb, together with 506 Tg yr⁻¹ isoprene and 126 Tg yr⁻¹ terpene emissions. The shaded regions indicate mid-latitudes. The assumed natural NOx emissions are 0.6 Tg N yr⁻¹ for stratosphere-troposphere exchange, 5.0 Tg N yr⁻¹ for lightning, 5.6 Tg N yr⁻¹ for soils and 6.8 Tg N yr⁻¹ for biomass burning.**





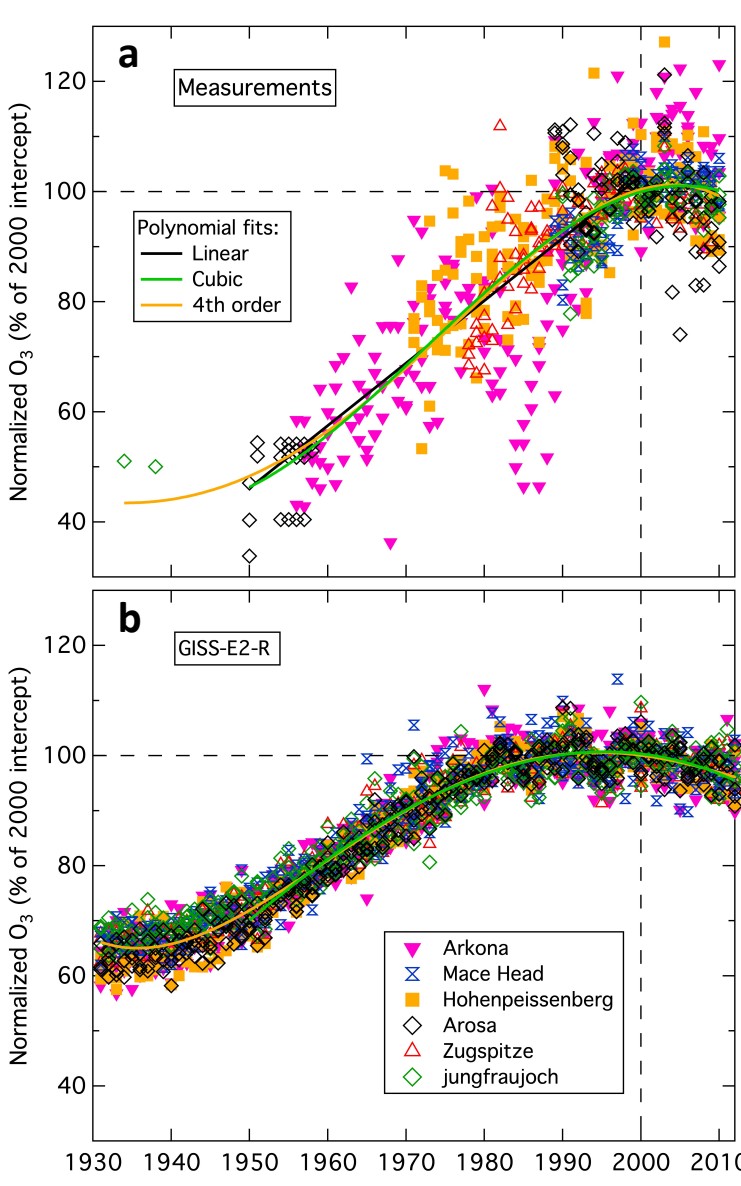

**Figure 2. Normalized, seasonal mean ozone measured (a) and simulated (b) at the six baseline representative European sites considered by Parrish et al. (2014). The simulations are from the GISS-E2-R model. Each graph includes cubic (beginning in 1950 - green curve) and 4th order polynomial (gold curve) fits to all seasonal means; (a) also includes a linear fit (1950-2000 – black line).**






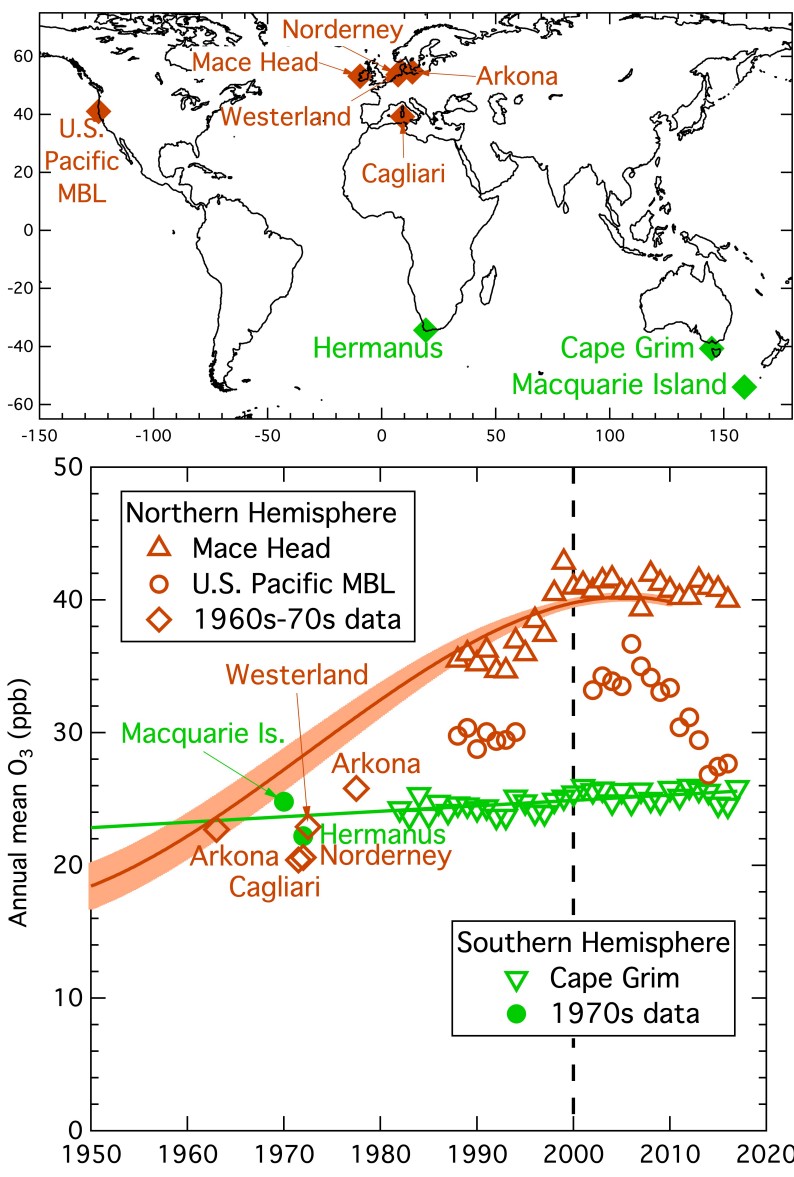

**Figure 3. Long-term changes in annual mean baseline ozone mixing ratios measured at mid-latitude, low elevation coastal sites indicated on the map in the northern (brown symbols) and southern (green symbols) hemispheres. The symbols for the longer-term, more recent data sets (Mace Head and Cape Grim) represent individual annual means; the symbols for the data from the 1960s and 1970s (Tarasick et al., 2019) represent averages over the variable periods (see Table 1) of available data. The derivation of the brown solid curve is described in the text; the shaded area about the curve indicates estimated confidence limits for the curve. The green solid line is a standard linear regression to the Cape Grim data, with extrapolation back to 1950.**



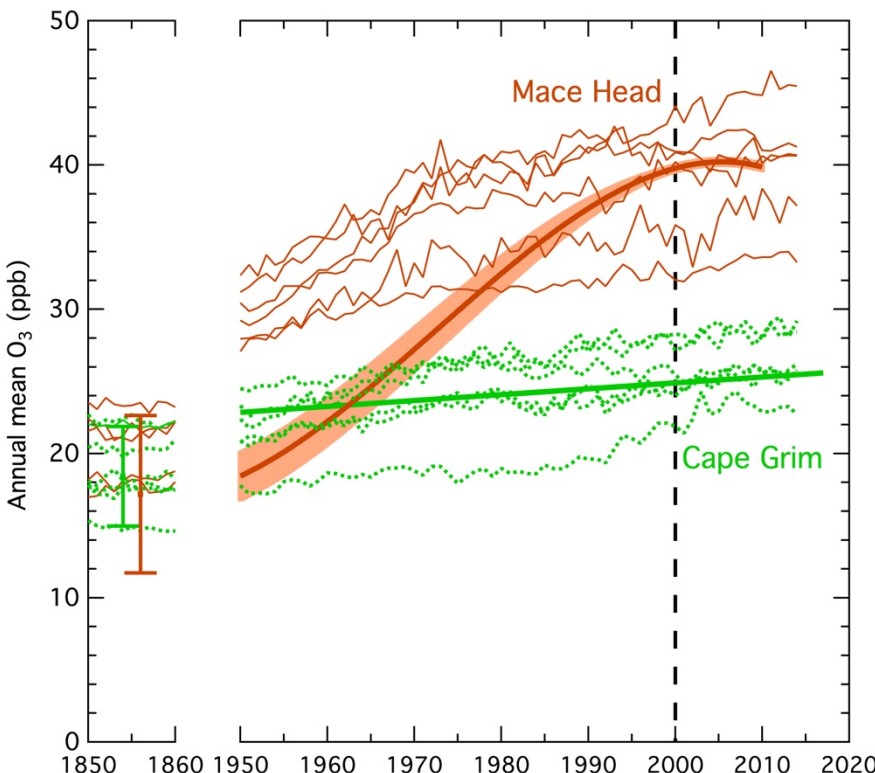

**Figure 4. Comparison of modeled and measured long-term changes in annual mean ozone mixing ratios at Mace Head and Cape Grim from 1850-1860 (left) and after 1950 (right). The fits to the measurements (heavy solid line and curve on the right) are the same as those in Figure 3. The thinner solid (for Mace Head) and dotted (for Cape Grim) curves are results from six ESMs that participated in the CMIP6 exercise. The two symbols on the left show the range of annual mean pre-industrial mid-latitude (30° to 60°) surface ozone calculated by the STOCHEM-CRI model for the 5 simulations discussed in the text.**



**Table 1.** Ozone data sets and analyses utilized in this work.

| Monitoring Site | Lat./Long. | Elev. (m) | Dates | Reference |
|---|---|---|---|---|
| | Southern Hemisphere | | | |
| Cape Grim, Australia | 40.7S, 144.7E | 100 | 1982–2017 | TOAR data base |
| Hermanus | 34.4S, 19.3E | 10 | 1970-1975 | Tarasick et al., 2019 |
| Macquarie Is. | 54.5S, 159.0E | 0 | 1970-1971 | Tarasick et al., 2019 |
| | Northern Hemisphere | | | |
| Mace Head, Ireland | 53.2N, 9.5W | 20 | 1988–2017 | Derwent et al., 2018a |
| Arkona | 54.7N, 13.4E | 40 | 1956-1984 | Tarasick et al., 2019 |
| Norderney | 53.7N, 7.2E | 20 | 1969-1975 | Tarasick et al., 2019 |
| Cagliari | 39.2N, 9.1E | 20 | 1970-1975 | Tarasick et al., 2019 |
| Westerland | 54.9N, 8.3E | 10 | 1971-1975 | Tarasick et al., 2019 |
| U.S. Pacific Coast MBL | 38-48N, 123-124W | 0-240 | 1988–2016 | This work |


**Table 2.** Coefficients of polynomials ($a + bt + ct^2 + dt^3$, where t = year - 2000) that define the long-term ozone changes given
by the cubic polynomial fit in Figure 2a and the solid curves in Figures 3 and 4.

| Western Europe* | a (%) | b (% yr$^{-1}$) | c ($10^{-2}$ % yr$^{-2}$) | d ($10^{-4}$ % yr$^{-3}$) |
|---|---|---|---|---|
| (all sites) | 100 ± 0.9 | 0.40± 0.11 | -3.4 ± 0.9 | -4.1 ± 1.8 |
| **Site** | **a (ppb)** | **b (ppb yr$^{-1}$)** | **c ($10^{-2}$ ppb yr$^{-2}$)** | **d ($10^{-4}$ ppb yr$^{-3}$)** |
| Cape Grim, Australia | 24.9 ± 0.2 | 0.041 ± 0.019 | --- | --- |
| Mace Head, Ireland | 39.8 ± 0.6 | 0.160 ± 0.043 | -1.35 ± 0.38 | -1.64 ± 0.73 |

* % unit indicates percentage of year 2000 intercept of annual means