# Peer review of "Investigations on the Anthropogenic Reversal of the Natural Ozone Gradient between Northern and Southern Mid-latitudes"

_Atmospheric Chemistry and Physics, 2020_

## Referee Comment (RC1)

**Review of "Anthropogenic Reversal of the Natural Ozone Gradient between Northern and Southern Mid-latitudes" (ACPD-2020-1198, Parrish et al)**

**General comment**
In my view, this is an excellent paper that combines observations and model outputs in a manner that illustrates the complex phenomena driving ozone changes since pre-industrial times, and highlights what should become a standard test for current Earth System Models aimed at estimating anthropogenic climate forcing. After minor revisions, I warmly recommend publication in ACP.

**Specific comments**
- In the text, there are several comparisons among trends derived from different observational and model output sets. Please specify the approach for calculating these trends and their corresponding error.
- Develop further the properties of the model STOCHEM-CRI that make it particularly useful for analyzing low NOx chemistry.
- Based on available long-term records in the Southern Hemisphere, your ague that the zonal variability of annual ozone means in the marine boundary layer is relatively small. However, Cape Grim, Ushuaia and Cape Point are subject to substantially different weather and (natural and anthropogenic) chemical regimes. Is this homogeneity also captured in models? What happens when you consider the seasonality of those data? Perhaps the similarity among ozone levels is just the result of compensating but quite different drivers.

---

## Author Comment (AC1)

**Reply to RC1**

*We are grateful for the referee's supportive comments. In the following we reproduce the original comments in* black regular font*, and include our responses in blue italic font. Revisions made to the manuscript are indicated in* **blue bold font.**

**General comment**

In my view, this is an excellent paper that combines observations and model outputs in a manner that illustrates the complex phenomena driving ozone changes since pre-industrial times, and highlights what should become a standard test for current Earth System Models aimed at estimating anthropogenic climate forcing. After minor revisions, I warmly recommend publication in ACP.

*Thank you for your supportive comments*

**Specific comments**

• In the text, there are several comparisons among trends derived from different observational and model output sets. Please specify the approach for calculating these trends and their corresponding error.

*Thank you for this suggestion.* **We have added the following text to the first paragraph of the Methods section:**

**Herein we discuss linear and polynomial fits to time series of ozone measurements; these fits with associated confidence limits are derived from standard least-squares fitting procedures, such as discussed in Chapters 6 and 7 of Bevington and Robinson (2003). All fits were performed with the Wavemetrics IGOR Pro software package.**

• Develop further the properties of the model STOCHEM-CRI that make it particularly useful for analyzing low NOx chemistry.

*And, thank you for this suggestion.* **The following text has been added to our revised manuscript:**

**The Condensed Reactive Intermediates (CRI) mechanism is a condensed version of the highly detailed and explicit Master Chemical Mechanism (MCM) v3.3.1 (https://mcm.york.ac.uk). Both rely entirely on evaluated laboratory chemical kinetics studies for their rate coefficient and product yield data, which is important in the present context of the pre-industrial atmosphere because low NOx conditions are not accessible in the smog chamber studies that have been an important source of mechanistic data for the chemistry of polluted atmospheres. Derwent et al. (2021) document the fidelity of the MCM and CRI mechanisms in a chemical mechanism intercomparison focused on the low NOx conditions of the pre-industrial troposphere.**

• Based on available long-term records in the Southern Hemisphere, you argue that the zonal variability of annual ozone means in the marine boundary layer is relatively small. However, Cape Grim, Ushuaia and Cape Point are subject to substantially different weather and (natural and anthropogenic) chemical regimes. Is this homogeneity also captured in models? What happens when you consider the seasonality of those data? Perhaps the similarity among ozone levels is just the result of compensating but quite different drivers.

*We agree that the three Southern Hemisphere sites are subject to substantially different weather and chemical regimes. However, at mid-latitudes in the free troposphere the lifetime of ozone (~3 months) is longer than either the mean circum-global transport time (~1 month in the prevailing westerly winds) or the average vertical overturning time (also ~1 month). This zonal and vertical transport implies that mean baseline ozone concentrations are relatively long-term and large spatial averages over the different ozone source and sink regimes throughout southern mid-latitudes. As a result, mean free tropospheric ozone concentrations and their seasonal changes are expected to be zonally similar. Entrainment of ozone from the free troposphere is the primary ozone source to the marine boundary layer (MBL) at the relatively remote baseline locations of these three sites. Consequently, the zonal similarity in the free troposphere is expected to be reflected in similar long-term changes and seasonal cycles at these MBL sites.*

*Parrish et al. (2016) compare measured and model simulated ozone concentrations at these three sites (see their Figure 4); the expected zonal similarity is seen in the measurements, and to a greater or lesser extent in simulations from three chemistry climate models. Figure 1 compares the seasonal cycles measured at the three sites (taken from Figure 4 of Parrish et al., 2016), and Figure 2 compares those same measurement curves with similar curves derived from the seasonal cycles calculated by the 6 ESMs discussed in our manuscript. These ESM simulations generally reproduce the qualitative features of the measurements (i.e., wintertime maximum and summertime minimum), but also exhibit some quantitative disagreements.*

*Overall, the zonal variability of both the annual ozone mean and the seasonal cycle in the marine boundary layer is relatively small in the observations.* **We now mention the similarity of seasonal cycles in the discussion of the small zonal variability of ozone in the marine boundary layer of the Southern Hemisphere.**

[Figure]

*Figure 1. Sum of fundamental and second harmonic fits to measured seasonal cycles of O₃ mixing ratios at the three southern hemisphere marine boundary layer sites identified in the annotation. Curves are taken from Figure 4 of Parrish et al. (2016).*

[Figure]

*Figure 2. Sum of fundamental and second harmonic fits to measured and modeled seasonal cycles of O₃ mixing ratios at the three southern hemisphere marine boundary layer sites included in Figure 1. Measurement curves are the same as in Figure 1. Model simulated seasonal cycle are from the 6 CMIP6 models discussed in our manuscript.*

---

## Author Comment (AC2)

**Reply to RC2 and RC3**

*Thank you for your time and effort that you put into reviewing our paper, and for your insightful comments, all of which are addressed below. In the following we reproduce the original comments in* black regular font*, and include our responses in blue italic font. Revisions made to the manuscript are indicated in* **blue bold font.**

This manuscript by Parrish et al. describes the analysis of trends in tropospheric ozone at mid-latitudes in both Northern and Southern Hemispheres over the last few decades, using the outputs from community efforts such as HTAP and TOAR. The manuscript is generally well written and presented, and provides a good account of the many factors affecting observed ozone concentrations, and of the challenges associated with modelling them. I have one major reservation on how the sites discussed are assumed to be representative of mid-latitudes for both hemispheres (see Major Comments below). There are also minor points (see below) that, once addressed, would make the text clearer. I recommend publication of this manuscript only once the comments below, especially the major ones, are suitably addressed.

**Major Comments**
My main reservation about the conclusions drawn by the authors of a reversal in the interhemispheric tropospheric ozone gradient in the pre-industrial era is that they mainly rely on the extrapolation of the fits to two series of observation from two background sites, one per hemisphere, and on the assumption that the polynomial fit derived from European monitoring sites (Fig. 2a) is representative of ozone trends for all NH mid-latitudes.
*This is an important comment. It is true that we rely on an extrapolation of the fit to the Cape Grim data to estimate past ozone changes in the SH; with regard to the uncertainty that results from this extrapolation, please see our response to a more extensive comment on this issue made by Cooper et al. (acp-2020-1198-CC1). However, it is important to note that* **we do not simply extrapolate the Mace Head data in the NH.** *Throughout northern mid-latitudes long-term changes in average baseline ozone concentrations are found to be the same, within narrow statistical confidence limits, when quantified in relative terms. This uniformity extends over all longitudes and through all altitudes from the surface to at least ~ 9km. The reasons for this uniformity and the evidence that supports it are discussed further in our responses following the next comment paragraph. Therefore, within a relatively small, quantified uncertainty we determine the past ozone changes that occurred at Mace Head before measurements were initiated, by normalizing that uniform relative change to the ozone concentrations measured at Mace Head from 1987-2017. The difference between a simple extrapolation and this normalization process may seem like a minor distinction, but it is of critical importance for accurately evaluating the uncertainty of our analysis.*

While the authors make a compelling case for their conclusions, for instance by showing how most European monitoring sites exhibit similar (relative) trends, it is also evident that some of the data shown in Fig. 2a (notably Arkona) exhibit deviations from the overall assumed trend. Furthermore, comparison with a non-European site (US Pacific MBL) shows potentially different temporal trends from its European counterparts (Fig 3). As the increase in tropospheric ozone in the NH is driven primarily by enhanced emissions of ozone precursors and nitrogen oxides (as the authors rightly point out), how do the authors justify assuming that these increases followed the same trend in Europe, North America and mid-latitude Asia throughout the time period considered here (1950-present)?
*This is another important comment. We do not simply* **assume** *that these increases followed the same trend throughout northern mid-latitudes;* **simple transport and ozone lifetime considerations, observations, and model simulations** *all support the* **conclusion** *that baseline ozone concentrations followed the same relative long-term changes throughout northern mid-latitudes. This allows us to confidently normalize the quantified relative long-term change to the recent Mace Head measurements. The observations and model simulations supporting this conclusion have been extensively discussed (HTAP, 2010; Parrish et al., 2012; 2014; 2020) and are illustrated in Figures 2 and S1 of our paper. Simple transport and ozone lifetime considerations support this conclusion; in the free troposphere at northern mid-latitudes the net lifetime of ozone is estimated as 100 days,*

*which is considerably longer than either the circum-global transport time (~30 days) or the vertical overturning time scale (~ 20 days). Consequently, even though the many sources and sinks of ozone are heterogeneously distributed, and each may possibly change over long time scales, the relatively rapid mixing and transport ensures that those changes are reflected in average baseline ozone concentration changes throughout northern mid-latitudes. In the presence of relatively rapid transport and mixing, there simply is no mechanism that can maintain heterogeneity in the long-term changes in the zonal baseline ozone concentrations. Parrish et al. (2020; 2021b) discuss these considerations in more detail.*

The authors convinced me that the sign of the ozone difference at Mace Head and Cape Grim (the two sites considered representative of their respective hemispheres) might have been reversed in the pre-industrial period for those two sites (or at least that their ratio might have converged to unity, as I have my reservations on the extrapolation of a 4th degree polynomial), however I'm not convinced that this can be extrapolated to all mid-latitudes based on these two sites alone. The authors point out that extrapolation to pre-1988 times (i.e., before the Mace Head record started) is apparently confirmed by measurements (Fig. 3), but as the authors point out, 3 out of 4 are from NH sites of questionable reliability, and the remaining site exhibited large variations in Fig 2 (Arkona). Why not show more points from the Arkona time series in Fig. 3? And perhaps the polynomial fit to the Arkona time series?

[Figure]

*Figure 1. Annual mean ozone mixing ratios measured at 4 low elevation coastal sites. Three sites with fits are taken from Figure 3 of our manuscript. The Arkona-Zingst data and fit are taken from Figure 1 of Parrish et al., 2021.*

*Figure 1 shows Figure 3 of our paper with the Arkona time series and its polynomial fit included. We do not include the Arkona data in Figure 3 of the paper, because it does not represent undisturbed MBL baseline air; it is a coastal site on the Baltic Sea that receives air with a strong continental influence, which reduces the mean ozone concentrations through surface deposition. The US Pacific MBL data are included to show that the Mace Head data are not abnormally low; in fact they are among the highest NH MBL baseline ozone concentrations observed.*

As for the SH, why was the Cape Point dataset not considered? It would be useful to include the Cape Point data series in Fig 3 as a term of comparison with the Cape Grim data, mirroring what the authors did for the NH data with Mace Head and the US Pacific MBL data.

[Figure]

*Figure 2. Annual mean ozone mixing ratios measured at 3 SH, low elevation coastal sites. The green line is a standard linear regression to the Cape Grim data, with extrapolation back to 1950. The 3 site mean with standard deviation is indicated in the annotation.*

*Figure 2 shows Figure 3 of our paper without the NH data, but with the 3 data sets that we considered for the MBL of the SH. Inclusion of the Cape Point and/or Ushuaia data sets in Figure 3 does not provide additional information as these data sets are generally consistent with the Cape Grim data as discussed by Cooper et al. (2014). Importantly, two approaches to estimating the preindustrial ozone concentrations at southern mid-latitudes (extrapolation of the Cape*

*Grim record to 1950, and simple averaging of all data reported from the three sites, under the assumption of no significant SH trend) give nearly identical results, as indicated in the figure annotations.*

I appreciate how some of the points raised here may be difficult to address due to the paucity of pre-1980 data, as the authors point out. However the paper would still make a valuable addition to the current literature on tropospheric ozone if the language in the discussion/conclusions section was adjusted to account for the sources of uncertainty in their analysis, as outlined below.
*In writing our paper, we attempted to carefully select language that struck the appropriate balance between clearly stating our quantitative conclusions and properly considering the uncertainty in our analysis. We appreciate the reviewer identifying specific instances where we failed to properly strike that balance. We respond to each of the reviewer's suggestions below.*

Line 20 (abstract): replace "likely" with "potentially"
*In Figures 3 and 4 of our paper, the shaded area about the NH curve indicates estimated confidence limits for the curve. That shaded area is entirely below the linear extrapolation of the SH line. The SH line is a conservative estimate of the past behavior (see our response to a comment included in the community comment acp-2020-1198-CC1). Thus, the NH curve is below the SH line well outside our estimated 95% confidence limits. Fully considering all of the issues involved, we believe that "likely" is the correct description in the full sentence on Line 20: "The available measurements indicate that this interhemispheric gradient was much smaller, and was likely reversed in the natural troposphere with higher concentrations in the SH."*

Line 221: replace "were" with "may have been"
*Following the same logic presented in the above response,* **we have replaced "were" with "likely were".**

Line 226: replace "must necessarily have been" with "might have been"
*Following the same logic presented in the two responses above,* **we have replaced "must necessarily have been" with "likely were".**

**Minor Comments**

line 22: replace "natural" with "pre-industrial"
**Replacement made**

line 34: add "However, tropospheric ozone…"
**"tropospheric" added**

line 47 (and again 53): I don't think "inconsistency" is the correct word in the context given. "aspect" would sound better in the paragraph in its current form. I'm assuming the authors think the findings of recent analyses (described in lines 47-55) are inconsistent with the NH always being thought of as the hemisphere with higher levels of pollutants?  I would suggest either replacing "inconsistency" or revising the paragraph.
**We have changed "inconsistency", and now refer to a "quantitative aspect" on line 47. On line 53, we have changed "resolve this inconsistency" to "explain this issue".**

Line 51 – add "higher in the NH than in the SH"
**Addition made**

Lines 145-150: Need to stress how this is strictly only valid across the temporal range for which measurements are available
**The final sentence on these lines has been modified to read: "Figure 2 indicates that to estimate the long-term ozone change at Mace Head (or any other baseline representative site in western Europe) over the 1950 to 2010 period for which measurements were analyzed, one needs only**

**to quantify the year 2000 mean ozone at the site, and then calculate the product of that intercept with the polynomial fit."**

Line 221: change to "Discussion and Conclusions"
**Change made**

Lines 222-226: You need to add the rate of increase in the SH for this sentence to make sense.
*Thank you for identifying this issue;* **this sentence has been revised to read: "First, the sparse measurement record at baseline sites indicates that between 1950 and 2000 ozone concentrations increased by a factor of 2.1 ± 0.2 in the NH (Figure 2 and Parrish et al., 2021) and a factor of 1.09 ± 0.04 in the SH (Cape Grim fit in Figure 3), which would imply an approximate doubling (factor of 1.93 ± 0.20); however present NH ozone concentrations are less than a factor of 2 greater than those in the SH (factor of 1.60 ± 0.03, based on year 2000 Mace Head to Cape Grim ratio), indicating that ozone concentrations likely were lower in the NH than the SH in 1950."**

Figures 3 and 4: Can you crop the y axis as starting from 10 ppb?
*One goal of these figures is to compare the absolute ozone concentrations between hemispheres in Figure 3 and between measurements and model simulations in Figure 4. Inclusion of the origin of the y-axis in the figures eases that comparison for readers, so we have not cropped the y-axis to start at 10 ppb.*

**Further comments I missed in my first review:**

Line 227: replace "were" with "may have been"
*Following the same logic presented in earlier responses above,* **we have replaced "were" with "likely were".**

Line 238: replace "were" with "may have been"
*The full sentence on this line reads: "Thus, although the measurement record is sparse, observations and the wider considerations discussed above are all consistent with the conclusion that in the preindustrial troposphere, mid-latitude ozone concentrations were higher in the SH than the NH." We believe that "were" is correct in this context, because we are correctly stating that "observations and the wider considerations" are consistent with a particular conclusion, not that the conclusion is necessarily true.*

*Additional References*

*Parrish, D.D., et al. (2020), Zonal similarity of long-term changes and seasonal cycles of baseline ozone at northern mid-latitudes. J. Geophys. Res.: Atmos., doi: 10.1029/2019JD031908. (updated reference – cited as 2020b in our original manuscript.)*

*Parrish, D.D., R.G. Derwent, and J. Staehelin (2021a), Long-term changes in northern mid-latitude tropospheric ozone concentrations: Synthesis of two recent analyses, Atmos. Environ., 248, https://doi.org/10.1016/j.atmosenv.2021.118227. (updated reference - cited as in review, 2020a in our original manuscript.)*

*Parrish, D.D., R.G. Derwent, and I.C. Faloona (2021b), Long-term baseline ozone changes in the Western US: A Synthesis of Analyses, Preprint https://www.essoar.org/doi/abs/10.1002/essoar.10506269.1*

---

## Author Comment (AC3)

**Comment on:**

**Anthropogenic Reversal of the Natural Ozone Gradient between Northern and Southern Mid-latitudes**

David D. Parrish, Richard G. Derwent, Steven T. Turnock, Fiona M. O'Connor, Johannes Staehelin, Susanne E. Bauer, Makoto Deushi, Naga Oshima, Kostas Tsigaridis, Tongwen Wu and Jie Zhang
* * *
**No Evidence for Anthropogenic Reversal of the Natural Ozone Gradient between Northern and Southern Mid-latitudes**

*Comment by Owen R. Cooper, David W. Tarasick, Ian E. Galbally and Martin G. Schultz*

The manuscript by Parrish et al. to which we refer with our comment misinterprets findings from the recent Tropospheric Ozone Assessment Report and draws conclusions that are not substantiated by the available data. Below we discuss the main issues that we have with the Parrish et al. study, as lead authors of the *TOAR-Observations* paper (Tarasick, Galbally et al., 2019).

*Thank you for your interest in our manuscript. Responding to the issues raised in your Comment allow us to further discuss some issues that evidently have caused some confusion. In the following we reproduce the original comment, which we refer to in our response as CTGS, in* black regular font*, and include our response in blue italic font. References are given in our manuscript or in CTGS, with some additional references given below.*

**Ozone increase since pre-industrial times:**

Interest in measuring and quantifying long-term changes in atmospheric composition grew slowly following the activities of the 1957 International Geophysical Year, and was brought into international focus by a meeting in Stockholm in 1971: a "Study of Man's Impact on Climate" (SMIC 1971). Subsequently Angell and Korshover (1983) produced the first systematic account of trends in tropospheric ozone in the 800-400 hPa layer, finding a 12% increase in ozone between 1970 and 1981. The notion that tropospheric ozone has increased greatly since pre-industrial times, however, seems to have originated with Bojkov (1986), based on an analysis of Schönbein papers, and the publication of trends in several decades of surface ozone measurements made at some sites in Germany (e.g. Arkona), by Feister and Warmbt (1987). Although more quantitative than the Schönbein papers, the Arkona measurements also used a difficult and obsolete chemical method with a large and uncertain bias relative to modern UV methods, and an additional bias due to SO2 interference. Nevertheless, with the then-recent recognition (Crutzen, 1973, 1974; Liu et al., 1980) that photochemistry could be a major source of tropospheric ozone, this hypothesis seemed likely, and was further reinforced by Volz and Kley's (1988) analysis of the data of Albert-Lévy (1877) from Montsouris. Many other analyses concluded that ozone in pre-industrial times was as low as 1/5 of its present concentration (e.g. Marenco et al., 1994; Sandroni et al., 1992; Sandroni and Anfossi, 1994; Pavelin et al., 1999).

These conclusions conflicted with those of some contemporary observers in the early 20th century, who had noted that surface ozone was typically about 20-25 ppb in clean air (Fabry, 1950), and also that the very low Montsouris measurements were not characteristic of rural air (Hartley, 1881). Moreover, global atmospheric chemistry models had difficulty reproducing such a large historical increase from pre-industrial times (e.g. Wang and Jacob, 1998; Mickley et al., 2001; Lamarque et al., 2005; Young et al., 2013; Parrish et al., 2014; Young et al., 2018). For these reasons *TOAR-Observations* (Tarasick, Galbally et al., 2019) set out to examine in detail the quality and credibility

of all known historical measurements of tropospheric ozone. This comprehensive reanalysis drew several conclusions:

1) There is no unambiguous evidence in the measurement record back to 1896 of typical mid-latitude background surface ozone values below about 20 ppb

2) As suggested by Fabry (1950), average values during the historical period for northern temperate regions were about 24 ppb

3) There is robust evidence both from the TOAR analyses of surface, balloon borne and aircraft observations (Tarasick, Galbally et al., 2019) and from the recent analysis of oxygen isotopes in ice-core data by Yeung et al. (2019) for increases in the northern hemisphere (NH) of 30-70%, with large uncertainty, between the period of historic observations, and the present day

4) The uncertainty in this estimated increase depends more on the modern region chosen for comparison than on the historical data, which are comparatively uniform. The representativeness of modern data thus seems to be the most important source of uncertainty in regional or global trend estimates.

*Thank you for giving this brief history of the investigation of long-term changes in tropospheric ozone. Some of the authors of our paper are old enough to fondly remember the 1970s and 1980s, when our understanding of atmospheric composition was just coming into focus. A meeting we particularly remember is the Workshop on Tropospheric Ozone held in Lillehammer, Norway in June 1987, where Crutzen (1988) gave an overview of the then current knowledge of tropospheric ozone. That overview provided a guide for much of the recent work that is germane to our present discussion. First, he collected much of the historical data that Tarasick, Galbally et al. (2019) evaluate at northern mid-latitudes, and concluded that "… at more remote background stations substantial ozone increases have taken place." Second, he recognized measurement uncertainties, e.g., he commented "It is clear that these very early measurements with the KI-technique must have been incorrect." and "the quality of some of the data may be doubtful." Third, he foreshadowed our own manuscript, noting that "Because ozone is much more efficiently destroyed on land surfaces than on the ocean, the fact that there is now so much more ozone in the northern hemisphere than in the southern hemisphere is indicative of substantial ozone production in the northern hemisphere." Finally, he concluded his review of trends in ozone concentrations with the suggestion that "it would be very interesting to compare certified old data with modern data taken at the same sites as where the "ancient" data were taken."*

*Notably, Crutzen (1988) did not quantitatively estimate the ozone changes. We attribute this to his concern regarding the reliability and paucity of the measurement records. Following his overview several efforts were made to make quantitative estimates; (e.g., Hough & Derwent, 1990). In our view, there have been two important developments since Crutzen's seminal work. First, the HTAP analysis (HTAP, 2010; Parrish et al., 2012; 2014) initiated Crutzen's closing suggestion. HTAP compared old data with modern data at the same sites to develop trend estimates, and importantly, noted that the same relative trends were found over large regions of northern mid-latitudes. Second, Tarasick, Galbally et al. (2019) effectively "certified" the old data. These two developments then allowed Parrish et al. (2021) to complete Crutzen's suggestion of comparing "certified old data with modern data taken at the same sites"; they derived a quantitative estimate of a factor of 2.1 ± 0.2 for the change in baseline ozone concentrations at northern mid- latitudes over the 1950–2000 period.*

*CTGS state four conclusions above. We judge that the first two are overly broad, and counter examples could be cited, but we do not have strong disagreements with either. We agree with the*

*4th conclusion that the representativeness of modern data is the most important source of uncertainty in trend estimates; indeed, this issue is key to the disagreement between our trend estimate of a factor of 2.1 ± 0.2 and those of Tarasick, Galbally et al. (2019), as thoroughly discussed in Parrish et al. (2021).*

*Finally, in our judgement, the 3rd conclusion is misleading if not clearly incorrect. The assertion of an increase of 30-70% would seem to disagree with our finding of an increase of 110 ± 20%, but that seeming disagreement may not be robust. As CTGS acknowledge, there is "large uncertainty" in the trend estimates of Tarasick, Galbally et al. (2019). This uncertainty arises because that work did not compare "certified old data with modern data taken at the same sites" as suggested by Crutzen (1988); rather that analysis compared old data with modern data from very different sites, a choice that led to two major problems. First, it was not clear to Tarasick, Galbally et al. (2019) how to compare the "old data with modern data"; they present a total of 30 different approaches to deriving trends (their Tables 7, S2 and S3), which gave estimated average trends varying between 32.0% and 70.8%, with 95% confidence limits that spanned the range of 9.3% to 95.9%; this upper confidence limit overlaps the lower confidence limit of our 110 ± 20% estimate. Second, and more importantly, their comparison of old data (collected at a set of primarily baseline representative sites at coastal and mountain locations) with modern data, collected at rural, low elevation sites within the European continental boundary layer, introduced several large biases into their comparison, all of which caused Tarasick, Galbally et al. (2019) to systematically underestimate the differences between the modern and old data. Parrish et al. (2021) discuss these biases in detail, and give estimates of their magnitudes. These magnitudes are large enough to easily account for the apparent factor of ~2 lower average trend estimate of Tarasick, Galbally et al. (2019) compared to our trend estimate of 110 ± 20%.*

*It should also be noted that Yeung et al. (2019) quantified changes in global tropospheric ozone, and derived an upper limit of 40% for the increase over the ~1950 to 2016 period. That limit is not in conflict with our northern mid-latitude increase estimate of 110%, since northern mid-latitudes account for only a small fraction (<18%) of the global troposphere, and it is recognized that ozone trends are much smaller in the rest of the troposphere compared to northern mid-latitudes (e.g., see Cooper et al., 2014).*

**What information can be gained from trends at a few sites in the center of Europe?**

While accepting the re-analyzed historical data presented in *TOAR-Observations*, the Parrish et al. manuscript concludes (along with, apparently, the paper cited as Parrish et al. 2020a), that ozone has increased in the NH by 110 ± 20%. Despite the surprisingly narrow confidence interval on this estimate, this conclusion is based on data from six sites: Jungfraujoch, Zugspitze, Arkona-Zingst, Mace Head, Hohenpeissenberg, and Arosa. Mace Head data were selected for "baseline" conditions, while the other sites contain available observations whose diurnal sampling is not explicitly stated. Arosa is a ski town at the bottom of a Swiss valley, Arkona-Zingst is a combined record from two sites on the north German coast, and Hohenpeissenberg is a hill in the middle of Germany. Jungfraujoch and Zugspitze are alpine sites and in daytime upslope winds bring air from the valleys to the summits. Cooper et al. (2020) showed that Jungfraujoch and Zugspitze are heavily influenced by the European boundary layer. The Jungfraujoch data before 1990 are very limited: 7 days in August 1933, and 5 days in August 1938. Older data from Arosa, Hohenpeissenberg and Zugspitze are from several wet chemical methods discussed by *TOAR-Observations,* and subject to $SO_2$ interference. Arkona data, as noted, are from the obsolete Cauer method and had significant biases relative to modern UV methods, rendering the Arkona-Zingst trend doubtful.

*The paper we originally cited as Parrish et al. (2020a) has now been published and is cited as Parrish et al. (2021). That paper did consider data from just the six European sites, because these are the only European data sets that we have identified where it is possible follow the suggestion of Crutzen (1988) "to compare certified old data with modern data taken at the same sites as where the "ancient" data were taken."*

*We do wish that data sets were available from more baseline sites and/or from sites more nearly completely isolated from recent continental influences, but we can only carefully analyze the data sets that do exist. Our analysis leads us to conclude that CTGS exaggerate the problems in the few existing data sets. The historical measurements from the Arosa area have been carefully evaluated (Staehelin et al., 1994). Cooper et al. (2020) include the Jungfraujoch and Zugspitze sites among the 20 "remote sites in the NH" that they consider; that paper does cite references that discuss the impact of anthropogenic sources on some atmospheric species at those sites; however, none of those references demonstrate that the ozone concentrations at these sites are directly impacted by the European boundary layer influence to a discernable extent. Importantly, Cooper et al. (2020) do not reference Chambers et al. (2016), who utilize measured radon concentrations as a tracer to unambiguously identify recent continental boundary layer (CBL) influence; that reference finds that ozone concentrations measured at Jungfraujoch do not vary with radon concentrations, which shows that the ozone concentrations are not significantly perturbed by CBL influences. Further, the Arkona data are not as questionable as CTGS suggest; notably Tarasick, Galbally et al. (2019) accept those data as suitable for historical reconstruction after a small correction for presumed $SO_2$ interference, and Parrish et al. (2021) present an analysis (their Section S4) that demonstrates that all northern German MBL data (their Figure S3) are generally consistent with the conclusion that ozone in the lower troposphere at northern mid-latitudes increased by a factor of 2.1 ± 0.2 from 1950 to 2000. More discussion of the influence of the CBL on baseline ozone concentrations is given below in response to another comment.*

Parrish et al. (2021) (like Parrish et al. 2020a) are in effect claiming that data from this set of six sites, despite their issues, are more representative of changes in the NH than the averages used by *TOAR-Observations*.

**We emphatically do indeed claim that the data from this set of six sites, with both certified old data and modern data collected at the same sites, are certainly far more representative of changes of northern mid-latitude baseline ozone concentrations than the comparisons presented in TOAR-Observations.** *The importance of this statement to this discussion cannot be overstated.*

This conclusion is inconsistent with recent studies of the spatial representativeness of surface ozone (cf. Sofen et al., 2016). *TOAR-Observations* averaged all available "rural" data over several large regions for its comparisons. Stations were classified as rural based on a set of stringent criteria including low population density (see Schultz et al., 2017) As can be seen from Figure 1, there are a large number of such sites in Europe and these are distributed across the continent. More than 250 rural sites in Europe were used in *TOAR-Observations*). The use of just a few sites, as a superior subset, requires justification, which is not provided in the Parrish et al. manuscript. Such a justification would have to consider in detail the representativeness of the modern data, since, as noted above (point 4), the differences between the TOAR and Parrish et al. estimates of the NH change lie in the different choices of modern data. If properly carried out, this would be an advance on the TOAR analysis. Data representativeness is a key issue for both air quality and climate models (cf. Sofen et al., 2016). Spatial representativeness is typically the largest source of uncertainty in the use of ground-based data; surface ozone is normally undersampled, as over land surfaces it may vary on scales of kilometres (Spangl et al., 2007). Current state-of-the-art model evaluation procedures attempt to represent these small-scale variations, and the changes/trends at many different places.

*The comments above are relevant to "rural" data collected at low elevations in the CBL, such as CTGS show in their Figure 1, so there is no inconsistency. In this environment, the effective ozone lifetime is quite short, so regional heterogeneity of trends is likely to be important. Such issues must be considered in the quantification of trends of local and regional ozone, such as that given by Chang, et al. (2017). However, in the baseline troposphere the effective ozone lifetime is long, yielding zonal similarity of long-term ozone changes (see more extensive discussion in our Reply on RC2 (https://doi.org/10.5194/acp-2020-1198-AC2)); the limited number of available data sets with both old data and modern data collected at the same site, not only allow effective quantification of baseline ozone changes at mid-latitudes, but that is the only approach of which we are aware that allows accurate quantification of these changes.*

*The TOAR database from surface sites, such as illustrated for Europe in Figure 1 of CTGS, is useful for a variety of investigations of ozone within the CBL; for example, impacts on human health (Fleming et al., 2018), on vegetation (Mills et al., 2018), and regional ozone trend analysis (Chang, et al., 2017) have been studied in some of the TOAR publications. However, that database is poorly suited for the investigation of larger-scale phenomena, such as average long-term baseline ozone changes that are the subject of our manuscript. This is because ozone concentrations at most continental surface sites are significantly perturbed by surface deposition and/or pollution ozone, photochemically produced from local precursor emissions or transported regionally. Fortunately, baseline ozone is a hemisphere-wide phenomenon that is zonally similar throughout northern mid-latitudes, which allow long-term changes to be accurately quantified from only the few available suitable data sets sites. This is the strategy that we employ in our manuscript, and its efficacy is supported by the references we cite.*

As noted above, five out of six sites used by Parrish et al. to determine baseline ozone trends are actually situated in the polluted European boundary layer. Figure 2 provides a simple illustration of the differences in both the phase and the amplitude of the annual cycles as well as the residual mean value between baseline ozone at Mace Head, Ireland and ozone at rural sites used in the Parrish et al. analysis. In particular, we emphasize the differences in behaviour of the sites of Zingst on the northern German coast and Hohenpeissenberg in southern Germany (both used by Parrish et al., 2021) to that of Mace Head.

[Figure]

**Figure 2**. Monthly median ozone, averaged over the years 2000-2015, at six rural sites in Western Europe. All data are from the TOAR database, with the exception of the baseline values at Mace Head, which are from Derwent et al. 2018.

The grey line shows unfiltered ozone at Mace Head Ireland, while the black line shows ozone filtered for baseline conditions (from Derwent et al., 2018). The baseline data show a peak in spring and a distinct summer minimum. Over the course of a year the average ozone value is 40.1 ppbv. Moving eastward we show the rural research station of Weybourne on the east coast of England (adjacent to the North Sea). The impact of emissions and depositional processes across Ireland and the UK are evident, as wintertime ozone at Weybourne is much less than at Mace Head, while summertime ozone is greater. Annual mean ozone is 6 ppbv less than at Mace Head, indicating a net loss of ozone. Two other sites on the North Sea coast (Norderney and Westerland) behave similarly to Weybourne. Zingst on the northern German coast also behaves similarly to the North Sea sites; annual mean ozone is 8 ppbv less than at Mace Head, again indicating a net loss of ozone. Hohenpeissenberg is quite different, showing a broad summer peak and a wintertime minimum, typical of polluted inland sites. Pollution influences at Hohenpeissenberg can also be seen from Figure 3, which shows the diurnal cycle of ozone mixing ratios at this site for different seasons averaged over two decadal periods. In particular, the range of the diurnal cycle is maximum in summer when photochemistry dominates.

**Hohenpeissenberg diurnal cycle of ozone mixing ratios**

[Figure]

**Figure 3**. Diurnal cycles of ozone mixing ratios at Hohenpeissenberg for different seasons and averaged over two decadal periods. All data are from the TOAR database. Note the different y-axis scales on the two panels.

Clearly, the interior sites behave differently from Mace Head and should not be used to infer trends of baseline ozone.

*Thank you for the interesting discussion of the seasonal and diurnal cycles at various European sites. However, we believe that CTGS have misinterpreted these cycles. Our Figure a) below illustrates the seasonal cycle from some of the same data sets, plus some additional data sets. All six of the baseline-representative data sets that we analyze are included. These seasonal cycles are derived from Fourier harmonic analysis of detrended monthly means as described in Parrish et al. (2019); the curves give the sum of the fundamental and second harmonics, which are generally the only significant harmonic contributors to the ozone seasonal cycle in baseline-representative data sets. The detrended data are normalized to year 2000, so they represent absolute ozone concentrations at somewhat earlier times than the data in Figs. 2 and 3 of CTGS. These curves represent the statistically significant information in the data sets regarding the seasonal cycles, without overfitting the data.*

[Figure]

*Figure a.* *Seasonal cycle in European baseline representative data sets. European sondes, MOZAIC, European alpine sites (Jungfraujoch, Zugspitze, and Sonnblick) and Mace Head curves are taken from Fig. 6c of Parrish et al. (2020). Arosa curve from Fig. 5 of Parrish et al. (2021). Hohenpeissenberg and Arkona-Zinst are derived from monthly mean data used in the Parrish et. al (2014), updated through 2017 at Hohenpeissenberg from monthly means from the TOAR database.*

*Our understanding of these seasonal cycles and the diurnal cycles of Fig. 3 can be briefly summarized:*

- *The three blue and green, higher altitude curves in Figure a) represent the seasonal cycle of baseline ozone in the free troposphere (FT). It is characterized by a late spring-early summer maximum and late autumn-early winter minimum. The agreement of the European alpine data with FT sonde and MOZAIZ aircraft data sets again emphasizes that the three European alpine sites do indeed represent the undisturbed FT baseline ozone concentrations with high fidelity.*

- *The Mace Head curve represents the seasonal cycle of baseline ozone in the marine boundary layer (MBL). It differs markedly from that in the FT, with a spring maximum and summer minimum. Entrainment of ozone from the FT is the primary source of ozone to the MBL, but the seasonal cycle is modified by rapid photolysis loss of ozone in the humid MBL. Parrish et al. (2016) give a detailed discussion of this process based on the Cape Grim MBL data. Note that MBL ozone is FT ozone that has been depleted by loss processes that have strong seasonal variations, but those loss processes are first order in ozone, so the long-term changes in MBL and FT ozone must exhibit the same **relative** long-term ozone changes. Thus, both MBL and FT baseline data can be used together in analysis of long-term changes, which is the approach used by Parrish et al. (2012; 2014; 2020).*

- *The curves from the three CBL sites (Arosa, Hohenpeissenberg, and Arkona-Zingst) in Figure a) correspond to FT baseline air entrained into the CBL with a variable admixture of MBL air at some sites. Ozone in the CBL also suffers first order loss (surface deposition). The overall influence of the variable mix of FT and MBL air and the depositional loss (which depends upon the local surface and meteorological conditions) likely account for the different mean ozone concentrations and the differences in the amplitude and phase of the seasonal cycles.*

- *The diurnal cycles that CTGS show for Hohenpeissenberg in their Figure 3 also result primarily from first order depositional loss to surfaces, so the same considerations apply.*

*We agree that the CBL sites behave differently from Mace Head and differently from the FT baseline data. However, this different behavior is primarily due to loss processes that are first order in ozone. Thus, FT, MBL and CBL data sets can all be analyzed together to infer **relative** long-term changes of baseline ozone. A possible concern that remains is the influence of local pollution ozone at the CBL sites, which could vary differently from baseline ozone on long-time scales. It is implicitly assumed that the local pollution influence on the derived long-term changes at the selected CBL sites is small;*

*this assumption is supported by the similarity of long-term ozone changes derived from all of the sites considered as discussed by Parrish et al. (2014; 2020).*

*In summary it is important to realize that a simple analysis of the seasonal or diurnal ozone cycle cannot be used to decide whether a particular site can be used to infer trends of baseline ozone, without a careful analysis of the cause of those cycles. It should be noted that a failure to account for the differences in the diurnal cycle between their historical and modern data led to one of the large biases in the Tarasick, Galbally et al. (2019) trend determination - see Section 3.6 of Parrish et al. (2021).*

**Unsupported hypothesis:**

The Parrish et al. paper is centered around the following hypothesis:

"…before the natural ozone distribution was perturbed by anthropogenic emissions of ozone precursors, the ozone gradient was reversed compared to that of today, with concentrations higher in the SH than the NH at mid-latitudes."

This is an intriguing suggestion for which there is some discussion in the literature of 40 years ago. However, there are no observations from the pre-industrial era that can support this hypothesis. There aren't even any data prior to 1970 in the mid-latitudes of the SH. Older data in the SH (Galbally and Roy, 1980; 1981) like those in Figure 3, show values indistinguishable from the historic averages for the NH.

The authors extrapolate the Cape Grim data back in time using standard linear regression. This is a doubtful exercise as there is no rationale to explain why ozone changes at Cape Grim should have been linear over extended periods of time prior to the beginning of the instrumental record. Parrish et al. then use their estimate of the NH trend to argue (their Figure 3) that this trend implies that ozone was lower in the NH in the pre-industrial era.

*It is true that no measurements are available before 1970 in the mid-latitudes of the SH; thus an extrapolation of some sort is necessary. We considered two limiting extrapolations: 1) the earliest measurements are assumed to represent the natural concentration, in which case that concentration would be assumed to remain constant back in time, and 2) the observed trend is assumed constant back in time. (These are limiting extrapolations, unless one speculates either that ozone concentrations were higher at times before the earliest measurements, or that the trend slope was even greater at earlier times; neither of these speculations have any theoretical or observational support, so are rejected.) The true trend before measurements began must lie between these two limits. We selected the more conservative limit for testing our hypothesis, i.e. that the SH ozone was lower in the past, so that NH ozone must have been even lower in the past to be lower than the extrapolated decreasing trend. Fortunately, both of this limits are consistent with our hypothesis.*

They ignore the fact that their extrapolation of the NH data (their Figure 3) puts the NH average well below the actual measurements (points 1&2, above). Because the models do not show this hypothesized reverse gradient, the authors conclude that the models are flawed. This conclusion, we are told, is necessary because "present NH ozone concentrations are less than a factor of 2 greater than those in the SH".

*As noted in our Reply on RC2 (https://doi.org/10.5194/acp-2020-1198-AC2), we are not extrapolating the Mace Head data; rather we are normalizing the quantified long-term changes of northern mid-latitude baseline ozone to the modern Mace Head data; the NH average of MBL ozone derived for 1950 in this manner is ~18 ppb, which is not well below the actual measurements that CTGS mention in their points 1&2 above). Notably, the model calculated MBL ozone concentrations in our Figure 3 in the mid-19$^{th}$ century in both hemispheres are at least qualitatively*

*consistent (14 to 24 ppb) with our estimate for the 1950 NH MBL concentration. We are not aware of any actual measurements from the MBL during preindustrial times that can compare with either our estimate or the model calculations.*

However, there is a simpler explanation for this alleged "inconsistency", namely that ozone has NOT increased in the NH by 110%, but by a smaller amount, as found by *TOAR-Observations*. Although they are not cited, the recent analysis of ice-core data by Yeung et al. (2019), and the independent analysis of aircraft and balloon data in *TOAR-Observations* both support a smaller increase of tropospheric ozone. We therefore conclude that there is presently no evidence to suggest that pre-industrial model simulations overestimate historic ozone levels, and we see no reason to criticize the models.

*As discussed in our response to the above first major comment by CTGS this simpler explanation is not acceptable, due to:*
*1) the large uncertainty and large biases in the TOAR-Observations approach as discussed in detail in Parrish et al. (2021).*
*2) the Yeung et al. (2019) analysis is consistent with our analysis as also discussed in that response.*
*3) the historical aircraft and balloon data are so uncertain and so sparse that they cannot support long-term tropospheric ozone change analysis before about 1998; see e.g., Logan et al. (1999; 2012).*

**Other Issues:**

1) Parrish et al. 2021 make the following claim about the Cape Grim ozone data:

"The annual mean Cape Grim data were downloaded from the TOAR data archive (Schultz et al., 2017; https://join.fz-juelich.de/access/, last accessed 20 April 2020); they were selected for baseline conditions as described in the TOAR data header."

This statement is incorrect. The TOAR Surface Ozone Database does not contain a baseline-selected subset of Cape Grim data, and there is nothing in the header information that would lead the user to such a conclusion.

*Thank you for this correction. The TOAR data header does have the statement:*

> *"accepted original data exclusions for short term events of hours to days"*

*We have worked with the PI of the Cape Grim ozone data on several analyses in the past, and are aware that a Cape Grim data set was prepared that was selected for baseline conditions. We mistakenly interpreted the quoted TOAR data header statement to mean that the exclusions were those necessary for baseline selection. We have changed the end of the sentence describing the Cape Grim data to read:*

> *"...; they were not selected for baseline conditions."*

2) The following statements by Parrish et al. (2020) can give the impression that the conclusions of *TOAR-Observations* support the findings of Parrish et al. (2021):

"As part of the Tropospheric Ozone Assessment Report https://igacproject.org/activities/TOAR), Tarasick et al. (2019) critically reviewed the record of historical ozone measurements throughout the global troposphere. Parrish et al. (2020a) have recently synthesized the HTAP and TOAR analyses."

"However, the accuracy of relative long-term ozone changes derived from these data is supported by the critical evaluation of Tarasick et al. (2019), which found no significant, systematic inaccuracy in the historical data analyzed by Parrish et al. (2012; 2014)."

As authors of *TOAR-Observations* we would like to clearly state that the conclusions of *TOAR-Observations* do not support the claim by Parrish et al. that baseline ozone has increased by a factor of 2.1 +/- 0.2.

*The TOAR-Observations authors are much too modest regarding the importance of their work; their efforts in evaluating the historical data have "certified old data" to the extent possible, which allowed Parrish et al. (2021) to quantify baseline ozone changes following the suggestion of Crutzen (1988): "it would be very interesting to compare certified old data with modern data taken at the same sites as where the "ancient" data were taken." We do rely on these efforts by the authors of TOAR-Observations.*

**Additional References**

*Chambers, S.D., Williams, A.G., Conen, F., Griffiths, A.D., Reimann, S., Steinbacher, M., et al., 2016. Towards a universal "baseline" characterisation of air masses for high-and low-altitude observing stations using Radon-222. Aerosol and Air Quality Research 16 (3), 885–899. https://doi.org/10.4209/aaqr.2015.06.0391.*

*Chang, K.-L., et al. (2017), Tropospheric Ozone Assessment Report: Present-day ozone distribution and trends relevant to human health, Elem Sci Anth, 5: 50, DOI: 10.1525/elementa.243*

*Crutzen, P.J., (1988) Tropospheric ozone: an overview, in: Tropospheric Ozone, edited by: Isaksen, I. S. A., D. Reidel Publishing Co., Dordrecht*

*Fleming, Z.L., et al. (2018), Tropospheric Ozone Assessment Report: Present-day ozone distribution and trends relevant to human health, Elem Sci Anth, 6: 12, DOI: 10.1525/elementa.73*

*Hough, A.M. & R.G. Derwent (1990), Changes in the global concentration of tropospheric ozone due to human activities, Nature, 344, 645-648*

*Logan, J.A., (1999) Trends in the vertical distribution of ozone: A comparison of two analyses of ozonesonde data, J. Geophys. Res., 104, 26,373-26,399.*

*Mills, G., et al. (2018), Tropospheric Ozone Assessment Report: Present-day tropospheric ozone distribution and trends relevant to vegetation, Elem Sci Anth. 6: 47, DOI:10.1525/elementa.302*

*Parrish, D. D., I. E. Galbally, et al. (2016), Seasonal cycles of $O_3$ in the marine boundary layer: Observation and model simulation comparisons, J. Geophys. Res. Atmos., 121, 538–557, doi:10.1002/2015JD024101.*

*Parrish, D.D., R.G. Derwent, S. O'Doherty, and P.G. Simmonds (2019), Flexible approach for quantifying average long-term changes and seasonal cycles of tropospheric trace species, Atmos. Meas. Tech., 12, 3383–3394, https://doi.org/10.5194/amt-12-3383-2019.*

*Parrish, D. D., Derwent, R. G., Steinbrecht, W., Stübi, R., Van Malderen, R., Steinbacher, M., et al. (2020) Zonal similarity of long-term changes and seasonal cycles of baseline ozone at northern midlatitudes. J. Geophys. Res.: Atmos., 125, e2019JD031908. https://doi.org/10.1029/2019JD031908.*

*Parrish, D. D, Derwent, R. G., and Staehelin, J., (2021), Long-term changes in northern mid-latitude tropospheric ozone concentrations: Synthesis of two recent analyses, Atmos. Environ., 248, https://doi.org/10.1016/j.atmosenv.2021.118227.*

Staehelin, J., Thudium, J., Buehler, R., Volz-Thomas, A. and Graber, W. (1994) Trends in surface ozone concentrations at Arosa (Switzerland). Atmos. Environ., 28, 75–87. doi: https://doi.org/10.1016/ 1352-2310(94)90024-8.

Wilson, S. R. (2015), Characterisation of $J(O^1D)$ at Cape Grim 2000–2005, Atmos. Chem. Phys., 15, 7337–7349, doi:10.5194/acp-15-7337-2015.

---

## Author Response (AR2)

*We greatly appreciate the editor's guidance and the reviewer's insightful comments. We have followed that guidance, and incorporated each of the suggestions into our revised manuscript. In the following we reproduce the original comments in* black regular font, *and include our responses in blue italic font. Revisions made to the manuscript are indicated in* **blue bold font.**

**Editor's Comments:**

Please make the changes suggested by the reviewer. Given that this is a contentious topic (it always has been!), it is really important that the points raised by Cooper et al are clearly addressed in the manuscript. In my opinion your work marks a significant step forward: however it is important not to overstate the case or it will be seen as just another contribution to a long-running debate. So please pay close attention to the wording of the statements.
**We have made all of the changes suggested by the reviewer; these changes addressed several of the points raised by Cooper et al. Additionally, the following sentence has been added to the penultimate paragraph of the Results section:**

**"Notably, Tarasick et al. (2019) conclude that baseline ozone increased in the NH by a smaller amount (30-70%, with large uncertainty) between the period of historic and present day observations; however Parrish et al. (2021a) show that their comparison of historic data (collected at a set of primarily baseline representative sites at coastal and mountain locations) with modern data, collected at rural, low elevation sites within the European continental boundary layer, introduced several biases into their comparison, all of which caused systematic underestimates in their derived differences between the historic and modern data. Parrish et al. (2021a) discuss these biases in detail and give estimates of their magnitudes, which are large enough to account for this apparent disagreement."**

*We think these changes address all of the major points raised by Cooper et al.*

*We have reviewed the wording throughout our manuscript, and are generally comfortable with the degree of certainty indicated, thanks in part to comments received in the first round of reviews.* **We have made three additional changes to ensure we do not overstate our case:**
- **Line 135: "… the most suitable method …" replaced with "… a suitable method …"**
- **Line 215: "We quantify the MBL baseline ozone mixing ratios as accurately as possible from these limited data, to allow a comparison of …" replaced with "We quantify the MBL baseline ozone mixing ratios as accurately as these limited data allow, in order to compare …"**
- **Line 289: The original wording: "The cause of the greater increases in the NH and reversal of the natural interhemispheric ozone gradient is attributed …" was qualified by adding "likely" before "reversal".**

**Reviewer's Comments:**

I would like to thank the authors for their thorough responses to my comments. Taking into account those as well as their reply to the comments by Cooper et al., I feel that some minor amendments are still needed to provide further context to the conclusions drawn as well as making the differences with other studies (e.g., TOAR) clearer to the reader.

In particular, there are very interesting sections in the author response that, if included in the main text, would enrich the manuscript greatly, by giving further elucidation on the underlying reasoning and making it more accessible to readers.

*Thank you for your careful reading and useful suggestions.*

More specifically, I would like the discussion on ozone lifetime to appear in some form in the main paper (start of page 5 on the collated Authors' Response, "Simple transport and ozone lifetime considerations support this conclusion; in the free troposphere at northern mid-latitudes the net lifetime of ozone is estimated as 100 days, which is considerably longer than either the circum-global transport time (~30 days) or the vertical overturning time scale (~ 20 days). Consequently, even though the many sources and sinks of ozone are heterogeneously distributed, and each may possibly change over long time scales, the relatively rapid mixing and transport ensures that those changes are reflected in average baseline ozone concentration changes throughout northern midlatitudes. In the presence of relatively rapid transport and mixing, there simply is no mechanism that can maintain heterogeneity in the long-term changes in the zonal baseline ozone concentrations. Parrish et al. (2020; 2021b) discuss these considerations in more detail"). This would lend confidence to the approach taken by the authors in analysing the relative trends in ozone (so it would fit nicely in the Methods section).

**We included this material as a separate paragraph in the Methods section:**

**"The HTAP-based analysis approach utilized here relies on the concept that baseline ozone concentrations followed the same relative long-term changes throughout northern mid-latitudes. Simple transport and ozone lifetime considerations support this picture; in the free troposphere at northern mid-latitudes the net lifetime of ozone is estimated as 100 days, which is considerably longer than either the circum-global transport time (~30 days) or the vertical overturning time scale (~ 20 days). Consequently, even though the many sources and sinks of ozone are heterogeneously distributed, and each can possibly change differently over long time scales, the relatively rapid mixing and transport ensure that those changes are all reflected in approximately constant average baseline ozone concentration changes throughout northern midlatitudes. In the presence of relatively rapid transport and mixing, there simply is no mechanism that can maintain heterogeneity in the long-term changes in the zonal baseline ozone concentrations. Parrish et al. (2020; 2021b) discuss these considerations in greater detail."**

I found the discussion on the differences between HTAP and TOAR in the response to the comments by Cooper et al. and in Parrish et al. 2021 very interesting, and I feel some of it needs to be included in the text to provide further context, perhaps in the shape of a brief summary on the findings of Parrish et al (2021) regarding the choice of sites to include in the analysis. This could appear somewhere in the long paragraph between lines 143 and 163 of the revised manuscript with track changes.

**We revised the first 3 sentences of the suggested paragraph in the Methods section to give a brief summary regarding the choice of sites that were included in the HTAP analysis:**

**"The HTAP analysis followed the suggestion of Crutzen (1988): "it would be very interesting to compare certified old data with modern data taken at the same sites as where the 'ancient' data were taken." The "ancient" data considered are the sparse record of early measurements made at baseline representative sites throughout Europe, which extend back to 1950, with two summer measurement periods from the 1930s. Tarasick et al. (2019) "certified" the "ancient data" to the extent possible by carefully evaluating the early measurements; these results were compared to modern data collected at those same sites."**

Lastly, I would recommend amending the title slightly, as in the absence of measurements from the pre-industrial era the anthropogenic reversal of mid-latitude ozone remains an intriguing hypothesis. Perhaps something along the lines of "Investigations on the anthropogenic reversal etc."

**We have changed the title as suggested:**

**"Investigations on the Anthropogenic Reversal of the Natural Ozone Gradient between Northern and Southern Mid-latitudes"**